# Queuine, a bacterial-derived hypermodified nucleobase, shows protection in *in vitro* models of neurodegeneration

Patricia Richard[1]*, Lucie Kozlowski[2], Hélène Guillorit[2,3], Patrice Garnier[2], Nicole C. McKnight[1], Antoine Danchin[4], Xavier Manière[2]*

1 Stellate Therapeutics Inc., JLABS @ NYC, New York, New York, United States of America, 2 Stellate Therapeutics SAS, Paris, France, 3 Institut de Génomique Fonctionnelle, Montpellier, France, 4 Kodikos Labs, Institut Cochin, Paris, France

* p.richard@stellate-tx.com (PR); x.maniere@stellate-tx.com (XM)

**Data Availability Statement:** All relevant data are within the paper and its Supporting Information files.

## Abstract

Growing evidence suggests that human gut bacteria, which comprise the microbiome, are linked to several neurodegenerative disorders. An imbalance in the bacterial population in the gut of Parkinson's disease (PD) and Alzheimer's disease (AD) patients has been detected in several studies. This dysbiosis very likely decreases or increases microbiome-derived molecules that are protective or detrimental, respectively, to the human body and those changes are communicated to the brain through the so-called 'gut-brain-axis'. The microbiome-derived molecule queuine is a hypermodified nucleobase enriched in the brain and is exclusively produced by bacteria and salvaged by humans through their gut epithelium. Queuine replaces guanine at the wobble position (position 34) of tRNAs with GUN anti-codons and promotes efficient cytoplasmic and mitochondrial mRNA translation. Queuine depletion leads to protein misfolding and activation of the endoplasmic reticulum stress and unfolded protein response pathways in mice and human cells. Protein aggregation and mito-chondrial impairment are often associated with neural dysfunction and neurodegeneration. To elucidate whether queuine could facilitate protein folding and prevent aggregation and mitochondrial defects that lead to proteinopathy, we tested the effect of chemically synthe-sized queuine, STL-101, in several *in vitro* models of neurodegeneration. After neurons were pretreated with STL-101 we observed a significant decrease in hyperphosphorylated alpha-synuclein, a marker of alpha-synuclein aggregation in a PD model of synucleinopathy, as well as a decrease in tau hyperphosphorylation in an acute and a chronic model of AD. Additionally, an associated increase in neuronal survival was found in cells pretreated with STL-101 in both AD models as well as in a neurotoxic model of PD. Measurement of queuine in the plasma of 180 neurologically healthy individuals suggests that healthy humans maintain protective levels of queuine. Our work has identified a new role for queuine in neuroprotection uncovering a therapeutic potential for STL-101 in neurological disorders.

**Funding:** The authors received no specific funding for this work. Authors [PR, LK, HG, PG, NCM, XM] are affiliated to Stellate Therapeutics. The funder provided support in the form of salaries for authors [PR, LK, HG, NCM, XM], but did not have any additional role in the study design, data collection and analysis, decision to publish, or preparation of the manuscript. The specific roles of these authors are articulated in the 'author contributions' section.

**Competing interests:** Authors [PR, LK, HG, PG, NCM, XM] are affiliated to Stellate Therapeutics. This affiliation does not alter our adherence to PLOS ONE policies on sharing data and materials.

## Introduction

Parkinson's and Alzheimer's diseases (PD and AD) are the most common neurodegenerative diseases, affecting seven and 44 million people worldwide respectively. PD is characterized by classical motor difficulties and non-motor manifestations that can occur years before diagnosis. Gastrointestinal symptoms such as constipation are among the main non-motor symptoms in PD that often occur a decade or more before onset [1]. Recently several studies showed that hallmarks of neurodegeneration such as Lewy bodies are found in the enteric nervous system (ENS) [2] and Parkinson's disease patients endure gut dysbiosis [3, 4]. Alzheimer's disease is the leading cause of dementia and is characterized by several clinical symptoms that include a progressive decline in memory, thinking, speech and learning capacities. Alzheimer's disease patients also show a change in their microbiota compared to healthy individuals [5–7]. Gut microbiota transfer in mouse models of PD and AD highlighted a functional link between bacteria composition and neurodegeneration [8, 9]. Altogether, a widespread understanding of these data is that communication between the gut and the brain, namely the gut-brain axis [10], is linked to disease.

Bacteria that have co-evolved with Eukaryotes are a source of many unique molecules that connect the gut with several organs including the brain, impacting human health and brain function. Queuine is a pyrrolopyrimidine-containing analog of guanine (Fig 1) that is exclusively synthesized by bacteria and found in most eukaryotes, including humans who acquire queuine from their own gut microbiota and a diet containing this bacterial-derived molecule [11]. Queuine produced by the gut microbiota is salvaged by humans through their gut epithelium and is distributed to a wide range of tissues including the brain where it is enriched [12]. Queuine is incorporated into the wobble position (nucleoside 34) of cytoplasmic and mitochondrial transfer RNAs (tRNAs) containing a $G_{34}U_{35}N_{36}$ (N = any nucleotide) anticodon sequence [13, 14]. In turn, its corresponding nucleoside queuosine is found in four specific tRNAs acceptors for the amino acids asparagine, aspartic acid, histidine and tyrosine (Fig 1) [11, 15, 16]. Queuine modification of tRNA (Q-tRNA) is an irreversible event [17] that occurs through a unique base-for-base exchange reaction (guanine replacement by queuine) by the eukaryotic heterodimeric tRNA guanine transglycosylase (TGT) enzyme composed of the catalytic QTRT1 and accessory QTRT2 subunits (Fig 1) [18]. Like most post-transcriptional modifications in the anticodon loop of tRNAs, especially position 34, queuosine is critical for the

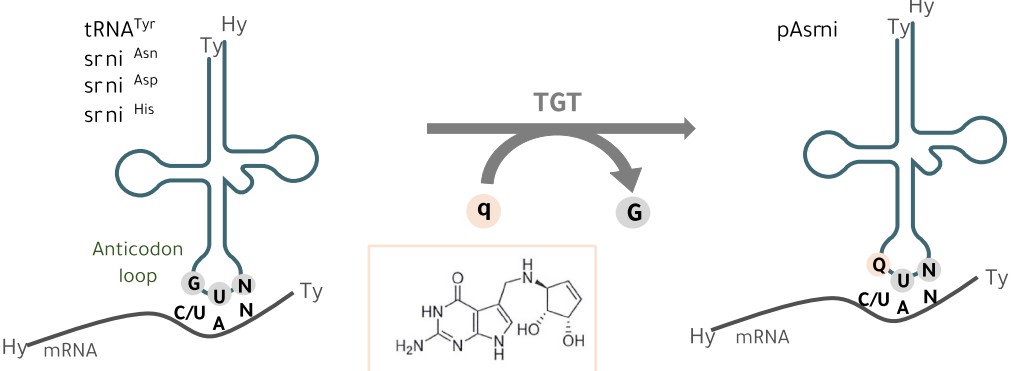

**Fig 1. Queuine integration into tRNAs.** In eukaryotic cells, the tRNA guanine transglycosylase (TGT) enzyme exchanges guanine (G) with the nucleobase queuine (q) at the wobble position (position 34, first base of the anticodon) of tRNAs that contain a GUN anticodon sequence (N = any base) and are specific to tRNA isoacceptors for tyrosine (tRNA^Tyr), asparagine (tRNA^Asn), aspartic acid (tRNA^Asp) and histidine (tRNA^His). Queuosine (Q) is the corresponding nucleoside of q and Q-tRNA base-pairing with NAY codons (Y = C or U) impacts speed and fidelity of mRNA translation [14, 20–22].

translation process [19]. Queuine incorporation into tRNA regulates translational speed and maintains accuracy [14, 20–22]. Deregulation of translation upon queuine depletion in both human and mice cells results in the formation of unfolded proteins that trigger endoplasmic reticulum (ER) stress and activation of the unfolded protein response (UPR) [14, 21]. Mitoribosome profiling in *QTRT2* KO cells recently showed that queuine at position 34 (Q34) controls elongation speed in mitochondria in a similar way than in the cytoplasm [14]. The 13 identified proteins translated in mitochondria are all components of the oxidative phosphorylation machinery [23] and a defect in mitochondrial protein homeostasis would dramatically compromise cellular metabolism. Mitochondrial dysfunction is indeed linked to several neurological disorders including PD and AD [24].

Another interesting link between queuine and PD is its requirement in maintaining a normal level of tetrahydrobiopterin (BH4), an essential co-factor for aromatic amino acid hydroxylases such phenylalanine hydroxylase (PAH) and tyrosine hydroxylase (TH). PAH synthesizes tyrosine from phenylalanine, which is then hydroxylated by TH to catalyze the formation of levodopa (L-DOPA), the precursor of dopamine [25]. PD is characterized by the death of dopaminergic neurons in the substantia nigra (SN), consequently leading to a dopamine deficiency that contributes to movement problems. While mice deprived of queuine appear asymptomatic, germ-free mice fed with a diet deficient in both tyrosine and queuine develop neurological abnormalities and die before 18 days [26]. Additionally, TGT-deficient mice have decreased levels of BH4 and elevated levels of its oxidized form BH2 (dihydrobiopterin), compromising tyrosine production, which indicates a crucial role for queuine in tyrosine metabolism [27]. BH4 is also a central co-factor for tryptophan hydroxylase (TPH) that leads to the production of serotonin, a neurotransmitter also deficient in PD patients [28]. While queuosine appears to block BH4 oxidation, queuine has also shown to protect HeLa cells against oxidative stress, another known contributor to neurological cell degeneration in PD [29, 30]. These findings imply that queuine plays an important role in dopamine synthesis through BH4 metabolism and prevention of oxidative stress.

Because queuine is enriched in the brain and promotes optimal cytoplasmic and mitochondrial mRNA translation necessary to prevent protein aggregation, a hallmark of several neurological disorders, we investigated the potential protective effect of synthesized queuine in several models of neurodegeneration.

## Results

### Synthesized queuine decreases alpha-synuclein hyperphosphorylation and is neuroprotective in a neurotoxin-based model of PD

Because queuine promotes optimal translation and gut microbiome changes have been linked to PD in several studies [4], we first investigated the effect of synthesized queuine (hereafter referred as STL-101) [31] in an *in vitro* model of synucleinopathy. Exposure of primary neuronal cells to recombinant human pre-formed alpha-synuclein (α-syn) fibrils (huPFFs) leads to the aggregation of endogenous α-syn into Lewy Bodies, a hallmark of PD [32]. We cultured wildtype (WT) primary mouse cortical neurons with or without STL-101 followed by huPFFs exposure for three weeks and monitored α-syn aggregation by immunofluorescence (IF) staining of α-syn phosphorylated at serine 129 (Ser129), a surrogate marker of *de novo* α-syn aggregation [33] (Fig 2). The Syn-211 antibody specifically detects human α-syn while the EP1536Y antibody specifically detects Ser129 phosphorylated form of α-syn (α-syn pSer129) of both human and rodent. After three weeks of huPFFs exposure, EP1536Y staining detected Lewy neurites and α-syn aggregates in the neuronal cell bodies of WT neurons exposed to huPFFs while only background signal was observed in control conditions (Fig 2A). Strikingly, STL-101

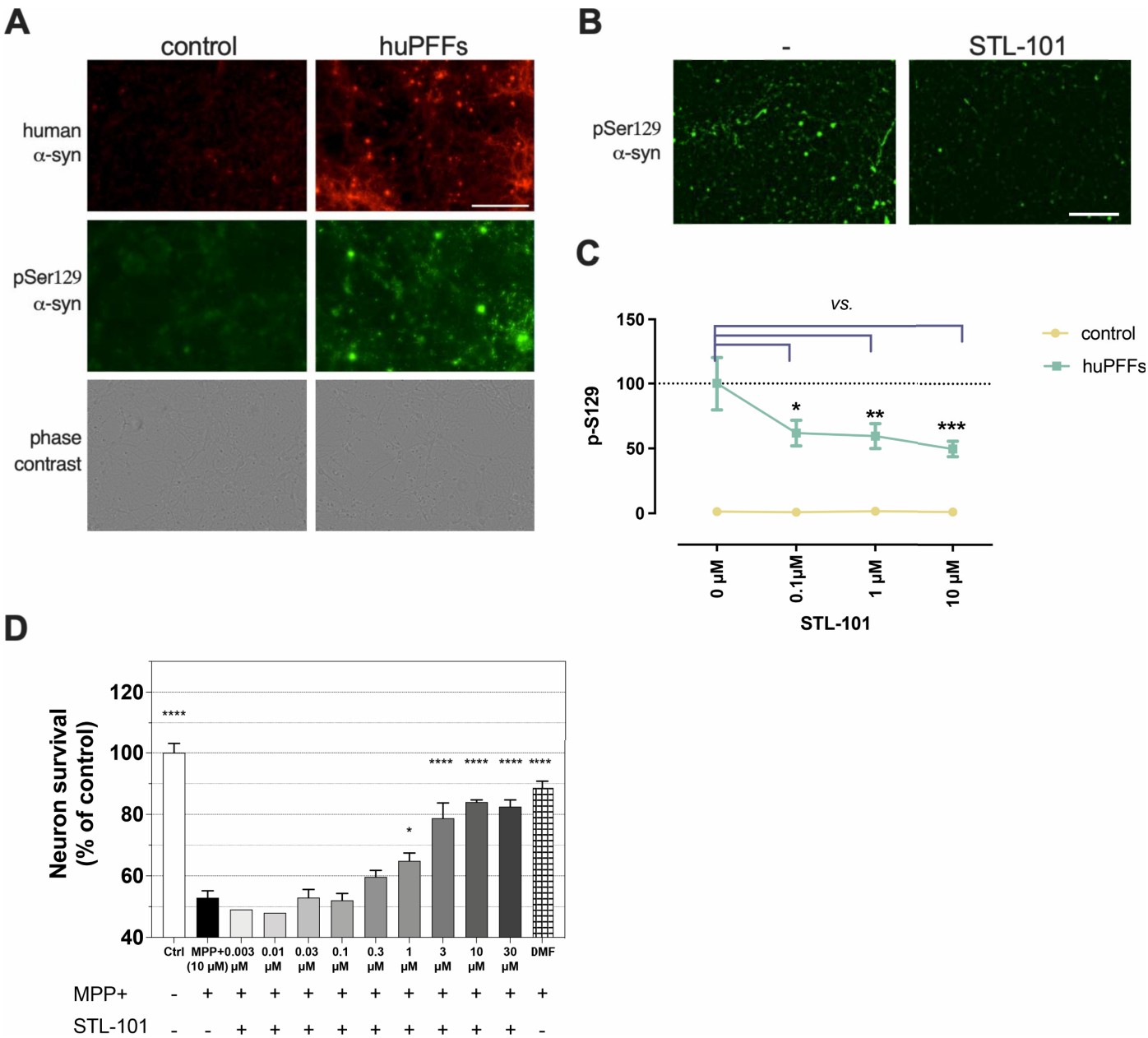

**Fig 2. STL-101 decreases α-syn pSer129 in an *in vitro* model of synucleinopathy and protects dopaminergic (DA) neurons from MPP+ injury. (A)** IF of human α-syn (labelled in red with syn211 antibody) and the phosphorylated/aggregated form of α-syn (α-syn pSer129, labelled in green with EP1536Y antibody) on control and huPFFs-treated mouse cortical neurons. Scale bar = 200μm. **(B)** IF of α-syn pSer129 (EP1536Y antibody) three weeks after huPFFs exposure in untreated (-) and treated mouse cortical neurons with STL-101 at 0.1μM. Scale bar = 200μm. **(C)** Quantification of α-syn pSer129 as shown in B. in control (no huPFFs) and huPFFs-exposed mouse cortical neurons treated with STL-101 diluted in PBS at 0, 0.1, 1 and 10μM. Data is expressed as mean +/- SEM and analyzed using two-way ANOVA followed by Dunnett's multiple comparison; ***p<0.001 cf. huPFFs + [STL-101] = 0μM. **(D)** STL-101 diluted in culture medium was added at the indicated concentrations to rat DA neuron cultures 1 day before MPP+ intoxication at 10μM. 48h after MPP+ injury DA neuron survival was assessed by TH+ neurons counting. Statistical significance was calculated using one-way ANOVA, Dunnett's multiple comparison test (*p<0.05, ****p<0.0001 in comparison to MPP + treatment only). n = 3 biological replicates. Dimethyl fumarate (DMF) was used at 10μM as a positive control.

pretreatment of huPFFs- exposed neurons led to significantly decreased α-syn pSer129, indicating a strong decrease in α-syn aggregation (Fig 2B). Quantification of EP1536Y antibody signal showed a ~40% decrease of α-syn pSer129 in cells pretreated with STL-101 at 100 nM

and up to ~50% in cells treated with 10μM of STL-101 compared to control cells not treated with STL-101 (Fig 2C).

In order to test the robustness of the effect of STL-101, we tested it in a different model of PD using the neurotoxin MPP+ (1-methyl-4-phenylpyridinium), which induces degeneration of dopaminergic (DA) neurons. MPP+ inhibits mitochondrial complex I leading to the production of reactive oxygen species (ROS) by mitochondria and subsequent death of DA neurons [34]. Rat DA neurons were cultured for six days *in vitro* (DIV) and treated with STL-101 at several concentrations (from 0.003μM to 30μM) one day before the addition of MPP+ at a final concentration of 10μM. Two days later, survival of DA neurons was assessed by counting TH+ neurons (Fig 2D). While MPP+ treatment resulted in 50% of DA neuron death, STL-101 treatment at 1μM and above significantly rescued DA neuron death leading to 65–85% survival of TH+ neurons. To confirm this result and test α-syn aggregation in the same model [35], we performed a similar experiment by co-treating rat DA neurons with MPP+ (at 4μM) and STL-101 at several concentrations (from 0.01μM to 10μM) (S1 Fig). Without pretreatment, STL-101 still showed a significant increase in neuronal survival at 3μM and 10μM after MPP+ intoxication (S1A Fig). It has previously been observed in this model that MPP+ injury triggers an increase in α-syn in DA neurons which was confirmed to be mainly aggregated α-syn [35]. Importantly, we measured a significant decrease of the α-syn signal detected by IF when the cells were treated with STL-101 at 1μM and above [~ 30% decrease compared to untreated cells (MPP+ alone)], indicating that STL-101 also impacts α-syn aggregation in an *in vitro* MPP+ model (S1B Fig).

These results show that STL-101 is neuroprotective and decreases α-syn aggregation in two distinct systems widely used to study PD, bringing to light its therapeutic potential in humans suffering from proteinopathy and neurodegeneration.

## STL-101 reduces tau hyperphosphorylation and cell death in *in vitro* models of AD

Alzheimer's disease is characterized by extracellular amyloid beta (Aβ) (~40 amino acid long peptide cleaved from the amyloid-β precursor protein (APP)) accumulation forming amyloid plaques, hyperphosphorylation and aggregation of tau forming intracellular neurofibrillary tangles (NFTs), and neuroinflammation. Increased levels of Aβ ($A\beta_{1-40}$ and $A\beta_{1-42}$ being the most abundant forms) can be found in the brain of AD patients and animal models and it is the Aβ oligomers that are believed to be the cause or at least contribute to the toxicity in AD pathogenesis [36]. AD is also associated with microglia activation, referred to as microgliosis, which is observed early in disease progression. First, in order to evaluate STL-101 capacity to reduce Aβ toxicity, we used an acute $A\beta_{1-42}$ oligomer injury model that reproduces essential neuropathological features of AD and a neuroinflammatory response [37]. Aβ peptides are produced as soluble monomers and undergo oligomerization with amyloid fibril formation via a nucleation-dependent polymerization process [38]. STL-101 was added to primary cultures of rat cortical neurons ten days after seeding and one day before $A\beta_{1-42}$ injury at 20 μM. 24 hours after $A\beta_{1-42}$ injury, IF staining of MAP-2 (Microtubule Associated Protein 2) and hyperphosphorylated tau (p-tau) was performed to evaluate survival of cortical neurons, neurite network and p-tau (Fig 3A). While $A\beta_{1-42}$ injury led to a significant increase in p-tau compared to control, treatment with STL-101 strongly reduced p-tau in a dose dependent manner (Fig 3B). STL-101 also protected the neurite network which is normally altered by $A\beta_{1-42}$ injury (Fig 3C). Importantly, pretreatment of STL-101 at a concentration of 30nM and above showed neuroprotection, increasing cell survival from approximately 70% to 80–85% at concentrations of 0.3μM and 1μM.

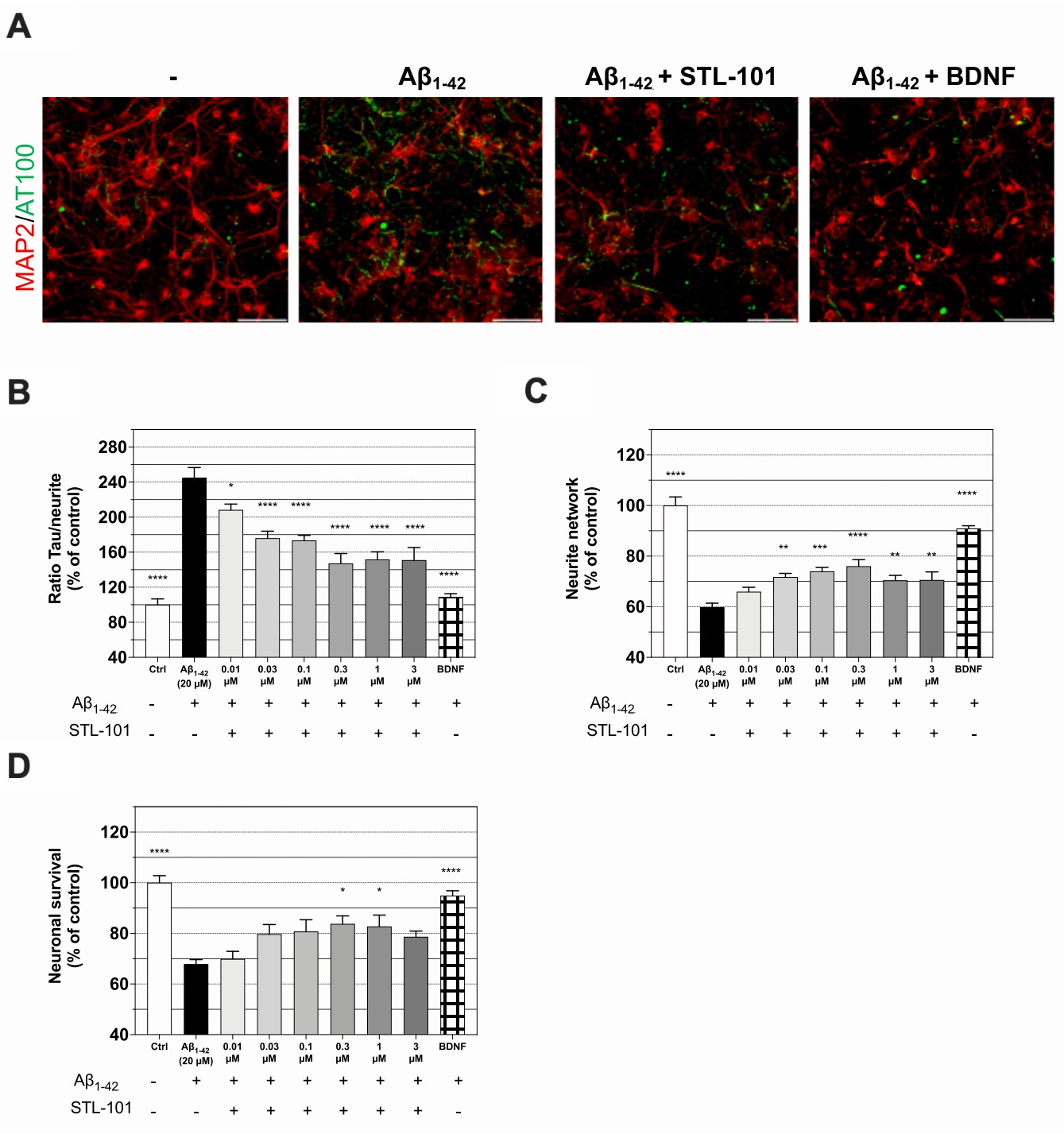

**Fig 3. STL-101 protects against acute injury with Aβ$_{1-42}$ in rat cortical neurons.** (A) Representative IF of MAP2 (red) and p-tau (AT100, green) in control (-), Aβ$_{1-42}$ injured neurons, Aβ$_{1-42}$ injured neurons pretreated with STL-101 at 0.3μM for 24h and Aβ$_{1-42}$ injured neurons treated with BDNF at 50ng/mL. Aβ$_{1-42}$ was used at 20μM. Scale bar = 100μm. (B) Quantification of p-tau in MAP2/AT100 IFs as shown in A. and with STL-101 pretreatment at the indicated concentrations. (C) Measurement of neurite network as shown in A. (D) Neuron survival quantification after Aβ$_{1-42}$ injury. STL-101 pretreatments were performed at the indicated concentrations and the neurotrophic factor BDNF was used as a positive control at 50ng/mL. Control (Ctrl) = culture medium with 0.1% DMSO. Results are expressed as a percentage of control condition and mean +/- SEM (n = 4–6 wells/condition) is shown. Statistical analysis was performed using one-way ANOVA followed by Dunnett's multiple comparison test (*p<0.05, **p<0.01, ***p<0.001, ****p<0.0001 in comparison with Aβ$_{1-42}$).

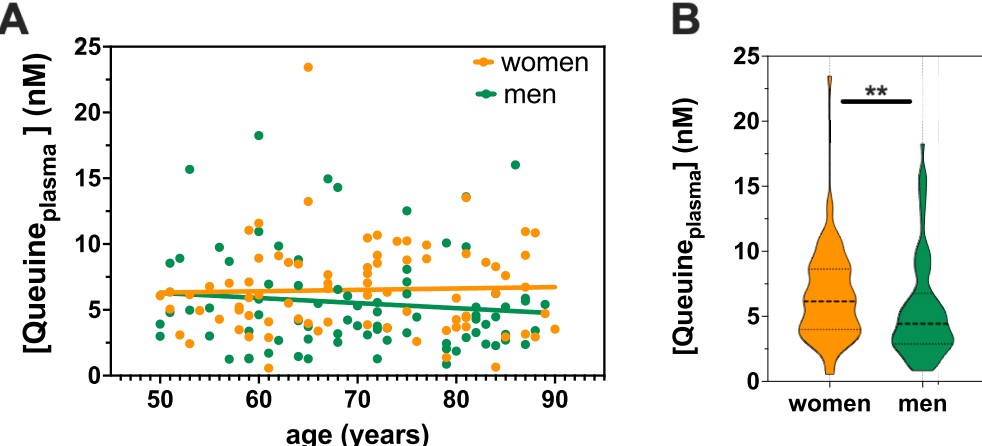

**Fig 4. Queuine level in plasma does not vary with age but is higher in women than men. (A)** Measurement of queuine level was performed by LC-MS/MS in plasma samples of 80 neurologically healthy women and 80 neurologically healthy men from 50 to 90 years old. **(B)** Comparison of queuine level between men and women as shown in A. $^{**}p<0.01$ (Mann Whitney test).

In addition, we tested the effect of STL-101 in both rat primary neuronal culture and microglia treated with $A\beta_{1-42}$ for 72 h at 5µM (S2 Fig). Similar to the effect after acute injury, chronic injury led to a significant increase in the neurite network at concentrations of STL-101 of 1µM and above (S1A Fig) and a decrease in p-tau at concentrations of STL-101 of 0.3µM and above (S1B Fig). We observed an associated neuroprotective effect of STL-101, however it was not statistically significant (S1C Fig). Of interest, TNF-α release in the culture medium induced by $A\beta_{1-42}$ injury was significantly lower in cells pretreated with STL-101 (at 0.3µM and above) compared to control (S1D Fig). These data indicate that STL-101 not only exhibits neuroprotective properties but also shows an anti-inflammatory effect in a chronic model of AD.

## Plasma queuine level is not age-dependent but higher in women

Neurodegenerative disorders are very often associated with aging. To determine any age-related or sex differences in the level of queuine, we measured queuine in the plasma of 80 neurologically healthy men and 80 neurologically healthy women from 50 to 90 years old. Measurement of queuine levels by LC-MS/MS showed no significant decline of queuine in plasma with increasing age (Fig 4A). This data is in perfect keeping with healthy individuals maintaining a protective queuine level while a decrease in queuine availability is likely occurring in patients experiencing dysbiosis. Unexpectedly, we found significantly higher level of queuine in the plasma of women compared to men (Fig 4B). Interestingly, analysis of the queuine levels in the plasma of 45 weeks old mice (~ 1 year old) also showed a higher level in female (> 3 times more) than male (S3 Fig).

## Discussion

Here we have shown protective properties of chemically synthesized queuine, a molecule naturally exclusively produced by bacteria, in several *in vitro* models of proteinopathies and neurodegeneration. We first showed that pretreatment of huPFFs-exposed neurons with chemically synthesized queuine (named STL-101 in this work), decreased α-syn pSer129 by approximately 50%. This indicates a strong reduction or prevention of α-syn aggregation normally triggered by exogenous huPFFs. We also showed a neuroprotective role of STL-101 in the

MPP+ neurotoxin model of PD and in both acute and chronic models of AD. STL-101 with or without pretreatment rescued about 15–35% of MPP+-exposed DA neurons from death and prevented/decreased α-syn aggregation by 30%. The neuroprotective effect identified in the acute AD model was associated with a decrease in phosphorylated tau and presumably the formation of NFTs, hallmarks of AD. Of importance, the chronic $A\beta_{1-42}$ injury model revealed a lower production of pro-inflammatory cytokine in STL-101-treated cells. We note that STL-101 concentration required for significant protection is higher with the MPP+ model than the other models (1μM vs. 0.1–0.3μM respectively). STL-101 was added 4 days prior huPFFs exposure, which might contribute to a better protection than no pretreatment or 1 day pretreatment before injury (MPP+ or $A\beta_{1-42}$). It is also possible that the mitochondrial stress induced by MPP+ alters STL-101 function (e.g., oxidation of the molecule, reduction of cellular intake) which would require higher amount of the molecule to reach protection. Interestingly, significant decrease of α-syn aggregation in the MPP+ model is seen when STL-101 is added without pretreatment at 1μM and above while significant neuronal survival is seen with STL-101 at 3μM. This indicates that some MPP+-induced phenotypes can be rescued by STL-101 near physiological concentrations (α-syn aggregation) while others require STL-101 at pharmacological concentrations (cell survival). This could also indicate that STL-101 treatment reduces/prevents α-syn aggregation before affecting cell survival. It is possible that by optimizing mRNA translation and preventing misfolding of newly synthesized protein, STL-101 delays and/or prevents protein aggregation (in a seeding model such as huPFFs as well as in a model where aggregation might be triggered by mitochondrial dysfunction (e.g., MPP+ intoxication)) that needs to reach a certain level before it can prevent neuronal loss. Together these data indicate that treatment with STL-101 offers several beneficial effects in various neurodegeneration models for which protein aggregates are a common feature. While we tested STL-101 as a preventive and co-treatment, it will next be important to test whether STL-101 offers similar benefits when administrated post injury.

Interestingly, we showed that the level of queuine does not vary with age in the plasma of healthy individuals and that the level is higher in women than men. It is well established that neurodegenerative diseases often show sex bias [39], for instance PD and amyotrophic lateral sclerosis (ALS) are more prevalent in men than in women while it is the opposite in AD and multiple sclerosis. It will be interesting to investigate whether these sex biases could be explained by the difference in queuine distribution and levels. This also highlights the importance of stratifying patients in order to select the subpopulation that would most likely benefit from specific therapy such as STL-101 treatment. Comparing queuine levels in biofluids and tissues of neurologically healthy individuals with patients affected by neurodegenerative disorders such as AD and PD will be of particular importance to substantiate our hypothesis that neurodegeneration is linked to queuine deficiency, possibly triggered by dysbiosis. Importantly, seeing similar sex difference in queuine concentration in the plasma of mice and humans indicate that mouse models will be suitable to further perform pharmacodynamics and pharmacokinetics studies.

Many disorders associated with defects in tRNA modification have been described and include metabolic diseases, respiratory defects, myopathies as well as several mitochondrial disorders [40]. On average, a tRNA contains 13 post-transcriptional modifications and a total of about 100 modifications have been identified in tRNAs, often in or adjacent to the wobble position that is crucial for mRNA decoding [41]. The human brain is highly sensitive to defects in tRNA modification and several neurological disabilities are linked to mutations in genes responsible for tRNA post-transcriptional modifications [42]. While a well-conserved and complex modification such as queuosine replacing guanosine could have been expected to

play important roles, its advantageous benefits and protective effects in neurons were unanticipated.

Another way for Q-tRNA to regulate mRNA translation is through its protection from cleavage into tRNA fragments that can interfere with translation. Indeed tRNAs lacking queuine modification are especially vulnerable to stress-induced cleavage by the ribonuclease angiogenin (ANG) in the anticodon loop [43, 44]. In fact, tRNAs undergo cleavage by several endonucleases leading to the production of a multitude of stable tRNA fragments (tRFs) of various sequences and sizes [45]. Among those tRFs, 5' tRNA halves and 3' tRNA halves (also named tRNA-derived stress-induced RNA or tiRNA), both products of ANG cleavage, are highly abundant under stress conditions [43, 46] and have been found to inhibit protein synthesis [47, 48]. Interestingly, tRNA halves have also been detected in the serum and CSF (cerebrospinal fluid) of PD patients and proposed to be used as biomarkers [49]. This role of queuine in tRNA cleavage inhibition and translation regulation can be expanded to its role in stimulating other tRNA modifications such as methylation of C38 in tRNA$^{Asp}$ as shown in *S. pombe* [50]. Similar to Q34, C38 has been shown to inhibit stress-induced tRNA cleavage by RNA endonucleases [44, 51]. Not only can queuine regulate stability of a pool of specific tRNAs but it can also contribute to protein synthesis inhibition through tRFs synthesis under stress condition [47, 48]. The protective role of queuine against tRNA cleavage is however in contrast with studies showing a neuroprotective role of ANG [52–54] and the fact that mutations in *ANG* have been linked to PD and ALS [55]. Nevertheless, ANG overexpression in the substantia nigra of mice model of PD slightly exacerbates dopamine neuronal death compared to controls [56].

While Q-tRNAs function in cytoplasmic and mitochondrial mRNA translation speed and fidelity, the mechanism leading to the protective effect on neurons is uncertain. TGT KO mice are viable and largely asymptomatic but do show symptoms similar to those associated with phenylketonuria (PKU) in that they are deficient in tyrosine production from phenylalanine and BH4 levels are significantly decreased in the plasma, while the BH4 oxidation product dihydrobiopterin (BH2) is elevated [27]. PKU results from a mutation in phenylalanine hydroxylase leading to phenylalanine buildup and associated neurological problems. While reduced BH4 levels have been linked to PD [25, 57], the exact mechanism through which queuine/queuosine impacts BH4 remains to be elucidated.

Aberrations in mitochondrial function can contribute to the molecular pathogenesis of both PD [58] and AD [59]. Mitochondria control translation of 13 subunits of the electron transport chain essential for oxidative phosphorylation [60] and the biogenesis of iron-sulfur cluster (ISC) essential for a broad range of cellular functions [61]. The mitochondrial unfolded protein response (UPR$^{mt}$) has been found to be activated in cortexes of AD patients and enhancement of mitochondrial function and proteostasis shows a reduction in Aβ aggregation and toxicity [62]. To potentially ensure proper translation of α-syn, tau or even APP, it is very likely that queuosine assures quality control of mitochondrial translation and homeostasis. A recent study found that queuine depletion promotes mitochondrial dysfunction resulting in an increased rate of proton leak and a decrease in ATP production in HeLa cells [63].

Queuine deficiency has also been linked to an increased progression of several human cancers [64]. Interestingly, an analog of queuine has already shown remarkable effect in an autoimmune disease mouse model of multiple sclerosis (EAE) by reversing disease hallmarks [65]. These reports and our work presented here demonstrate the therapeutic potential of chemically synthesized queuine (STL-101) possibly as a director of both general and specific mRNA translation quality control and may play a crucial role in optimal brain function and in protection against proteinopathies in particular. To confirm this hypothesis and further understand

the molecular mechanisms behind such function, we will continue experiments with STL-101 including *in vivo* mouse models of neurodegeneration.

## Materials and methods

### STL-101 synthesis

STL-101 has been synthesized by Synthenova SAS following the protocol established by Brooks et al. [31]. STL-101 is queuine dihydrochloride ($C_{12}H_{15}N_5O_3$ (2HCl)) with a molecular weight (MW) of 350.20 g.mol$^{-1}$ (natural queuine has a MW of 277.28 g.mol$^{-1}$). STL-101 purity was determined at 99% by HPLC and was solubilized in water, PBS or DMSO at 10–30 mM as a stock solution. Spectra related to STL-101 analysis can be found in the S1 Appendix supporting information file.

### Primary culture of cortical neurons and huPFFs challenge

All procedures related to this part of the study have been conducted in accordance with the European Communities Council Directive (2010/63/EU) for care of laboratory animals, with approval from the Institutional Care and Ethical Committee of Bordeaux University (CE50, France) under license number 20835. Timed pregnant female mice were received from Charles River Laboratories 2 days before initiation of the primary culture and sacrificed by cervical dislocation. Cortices were harvested from the E18 embryos of WT mice and dissociated enzymatically and mechanically (using the neuronal tissue dissociation kit, C-Tubes and an Octodissociator with heaters, Miltenyi Biotech, Bergisch Gladbach, Germany) to yield a homogenous cell suspension. 20,000 cells were plated per well in poly-D-Lysine-coated 96-well plates in a neuronal medium containing 0.5% Penicillin/Streptomycin and 0.5mM L-glutamine. The cultures were incubated at 37˚C / 5% CO2. At DIV 3 they were exposed to STL-101 at 3 concentrations (10μM, 1μM, 0.1μM in PBS) or none (PBS only). STL-101 treatment was then renewed every 3 days (by replacing 1/3 of the volume of medium with fresh medium supplemented with 1X concentration of STL-101). At DIV 7, half of the culture were exposed to huPFFs (at 1 concentration, i.e. equivalent to 10nM α-syn monomer). At 30 DIV, the neuronal cultures were fixed with PFA, and the effects of the treatments were evaluated by performing two double immunostainings: (i) D37A6/Syn1 to detect total rodent α-syn/human plus rodent non aggregated α-syn. (ii) EP1536Y/Syn211 to detect rodent and human α-syn phosphorylated at S129/total human α-syn. The induction of α-syn phosphorylation was quantified by measuring the accumulation of phospho-α-synuclein (length of the phospho-α-synuclein-positive neurite network, number of phospho-α-synuclein positive cell bodies). The experimental conditions were performed in quadruplicate (for STL-101-treated conditions +/-huPFFs) and triplicate (for untreated +/- huPFFs) wells. An array of nine individual microscopic fields were acquired in each single well, and 3 channels were recorded for each field (green fluorescence excited @488 nm, red fluorescence excited @594nm, and phase contrast).

### Culture of mesencephalic dopaminergic neurons, STL-101 pretreatment and MPP+ injury at 10μM and immunostaining

Animals were treated in accordance with the Guide for the Care and Use of Laboratory Animals (National Research Council, 1996), European Directive 86/609, and the guidelines of the local institutional animal care and use committee. Animals were sacrificed by cervical dislocation and cell cultures were prepared from the ventral mesencephalon of 15.5 days Wistar rat embryos (Janvier Breeding Center, Le Genest St Isle, France). Dissociated cells in suspension obtained by mechanical trituration of midbrain tissue pieces were seeded at a density of 1.2–

1.5 $10^5$ cells/cm$^2$ onto tissue culture supports pre-coated with 1mg/mL polyethylenimine diluted in borate buffer pH 8.3 as described in Michel et al., 1997. The cultures were maintained in Neurobasal medium supplemented with B27, 2mM L-glutamine, 100mg/ml streptomycin, and 100U/ml penicillin as well as K$^+$ (30mM) and MK-801 (5μM) to avoid excitotoxicity. Note that tyrosine hydroxylase (TH$^+$) neurons represent approximately 1–2% of the total number of neuronal cells present in these cultures.

STL-101 (culture medium was used as vehicle) was added at DIV6 and 1 day before MPP + addition at 10μM. 48 hours following MPP$^+$ treatment, cultures were fixed for 12 min using 4% formaldehyde, then washed twice with PBS before an incubation step at 4˚C for 24–48 h with an anti-TH antibody produced in mouse and diluted at 1/5000 (anticorps-enligne, #ABIN617906) to assess the number of DA neurons; Detection of anti-TH antibody was performed with a secondary anti-mouse CF-488 antibody (1/500, Ozyme, #BTM20208). TH$^+$ neurons counting was performed at ×200 using a ×20 objective matched with a ×10 ocular. The number of TH$^+$ neurons in each culture well was estimated after counting 6 visual fields distributed along the *x* and *y* axes.

## Culture of mesencephalic dopaminergic neurons, STL-101 and MPP+ (4μM) co-treatment, immunostaining and analysis

Experiments were carried out in accordance with the National Institutes of Health Guide for the Care and Use of Laboratory Animals and followed current European Union regulations (Directive 2010/63/EU). Agreement number: B1301310. Rat dopaminergic neurons were cultured as described by Visanji et al. [66], 2008 and Callizot et al., 2019 [35]. Briefly, pregnant female rat (Wistar) of 15 days of gestation were killed using a deep anesthesia with CO2 chamber and a cervical dislocation. The midbrains obtained from 15-day-old rat embryos (Janvier, France) were dissected under a microscope. Embryonic midbrains were removed and placed in ice-cold medium of Leibovitz (L15) containing 2% of Penicillin-Streptomycin (PS) and 1% of bovine serum albumin (BSA). The ventral portion of the mesencephalic flexure, a region of the developing brain rich in dopaminergic neurons, was used for the cell preparations. Midbrains were dissociated by trypsinization for 20 min at 37˚C (solution at a final concentration of 0.05% trypsin and 0.02% EDTA). The reaction was stopped by the addition of DMEM containing DNAse I grade II (0.5 mg/mL) and 10% of fetal calf serum (FCS). Cells were then mechanically dissociated by 3 passages through a 10 ml pipette. Cells were then centrifuged at 180 x g for 10 min at +4˚C on a layer of BSA (3.5%) in L15 medium. The supernatant was discarded, and cell pellets were re-suspended in a defined culture medium consisting of Neurobasal supplemented with B27 (2%), L-glutamine (2 mM) and 2% of PS solution and 10 ng/mL of Brain-derived neurotrophic factor (BDNF) and 1 ng/mL of Glial-Derived Neurotrophic Factor (GDNF). Viable cells were counted in a Neubauer cytometer using the trypan blue exclusion test. The cells were seeded at a density of 40,000 cells/well in 96 well-plates (pre-coated with poly-L-lysine) and maintained in a humidified incubator at 37˚C in 5% CO2/95% air atmosphere. Half of the medium was changed every 2 days with fresh medium.

On day 6 of culture MPP+ was added at a final concentration of 4μM at the same time than STL-101 diluted in culture medium and added at the final concentration of 0.01, 0.03, 0.1, 0.3, 1, 3 or 10μM. BDNF was added 1h prior MPP+ exposure. Cells were fixed 48h later with 4% PFA for 20 min at room temperature (RT) before permeabilization. Non-specific sites were blocked with a solution of PBS containing 0.1% of saponin and 1% FCS for 15 min at RT. Immunostaining was performed with anti-Tyrosine Hydroxylase (TH) (Sigma, #T1299, 1/10,000) and anti-α-syn (Cell signaling, 2642S, 1/200) antibodies for 2h at RT. Alexa Fluor 488

goat anti-mouse IgG (Sigma, #SAB4600042, 1/800) and Alexa Fluor 568 goat anti-rabbit IgG (Sigma, #SAB4600084, 1/400) were respectively used as secondary antibodies for 1h at RT.

Analysis of total number of TH neurons (TH+ neurons), total neurite network of TH+ neurons (in μm) and overlapping between TH and α-syn staining has been performed as followed: For each condition, 20 images (representing the whole well area) were automatically taken using ImageXpress® (Molecular Devices) at 10x magnification (20 pictures, for TH and α-syn signals into TH+ neurons) using the same acquisition parameters. From images, analyses have been automatically performed by Custom Module Editor® (Molecular Devices).

## Culture of cortical neurons for Aβ$_{1-42}$ acute exposure

The following procedure was carried out in accordance with the National Institutes of Health Guide for the Care and Use of Laboratory Animals and followed current European Union regulations (Directive 2010/63/EU). Agreement number: A1301310. Rat cortical neurons were cultured as described by Callizot et al., 2013 [37]. Briefly pregnant female rat (Wistar) of 15 days of gestation was deeply anesthetized with $CO_2$ and then sacrificed by cervical dislocation. Fetuses were collected and immediately placed in ice-cold L15 Leibovitz medium with a 2% penicillin (10,000U/mL) and streptomycin (10mg/ml) solution (PS) and 1% bovine serum albumin (BSA). Cortices were treated for 20 min at 37˚C with a trypsin-EDTA solution at a final concentration of 0.05% trypsin and 0.02% EDTA. The dissociation was stopped by addition of Dulbecco's modified Eagle's medium (DMEM) with 4.5g/L of glucose, containing DNAse I grade II (final concentration 0.5mg/mL) and 10% fetal calf serum (FCS). Cells were mechanically dissociated by three forced passages through the tip of a 10mL pipette. Cells were then centrifuged at 515 x g for 10 min at 4˚C. The supernatant was discarded, and the pellet was resuspended in a defined culture medium consisting of Neurobasal medium with a 2% solution of B27 supplement, 2mmol/L of L-glutamine, 2% of PS solution, and 50ng/mL of brain-derived neurotrophic factor (BDNF). Viable cells were counted in a Neubauer cytometer, using the trypan blue exclusion test. The cells were seeded at a density of 25,000 per well in 96-well plates precoated with poly-L-lysine and were cultured at 37˚C in an air (95%)—$CO_2$ (5%) incubator. For 96 wells plates, only 60 wells were used. The wells of first and last lines and columns were not used (to avoid the edge effect) and were filled with sterile water. The medium was changed every 2 days.

## STL-101 treatment and human Aβ$_{1-42}$ exposure

On day 10 of culture, STL-101 was dissolved in the culture medium (final concentration of DMSO = 0.1%) and pre-incubated with primary cortical neurons for 24 hours, before Aβ$_{1-42}$ exposure. STL-101 was tested at 0.01, 0.03, 0.1, 0.3, 1 and 3μM. Then, on day 11 of culture, the cortical neurons were injured with Aβ solutions. The Aβ$_{1-42}$ preparation was done following the procedure described by Callizot et al., 2013. The batch number of the Aβ$_{1-42}$ was used in this study contains 10% of oligomers (quantification by WB). Briefly, Aβ$_{1-42}$ peptide was dissolved in the defined culture medium mentioned above, at an initial concentration of 40μmol/L. This solution was gently agitated for 3 days at 37˚C in the dark and immediately used after being properly diluted in culture medium to the concentration used (20μM, 2μM oligomers evaluated by WB). Aβ$_{1-42}$ preparation was added to a final concentration of 20μM (2μM oligomers, AβO) diluted in control medium in presence of the compound, for 24 hours.

## Immunostaining after Aβ$_{1-42}$ exposure

24 hours after Aβ application, the cell culture supernatant was removed and discarded, and the cortical neurons were fixed by a cold solution of ethanol (95%) and acetic acid (5%) for 5 min

at -20˚C. The cells were washed twice in PBS, permeabilized, and non-specific sites were blocked with a solution of PBS containing 0.1% of saponin and 1% FCS for 15 min at room temperature.

After permeabilization with 0.1% of saponin, cells were incubated for 2 hours with: (i) chicken polyclonal antibody anti microtubule-associated-protein 2 (MAP-2) (Abcam, #Ab5392) at dilution of 1/1000 in PBS containing 1% fetal calf serum and 0.1% of saponin (this antibody stains specifically cell bodies and neurites, allowing study of neuronal cell death and neurite network). (ii) mouse monoclonal anti-hyperphosphorylated tau (Thr212, Ser214; AT100) (Fisher Scientific, #11818711) at the dilution of 1/100 in PBS containing 1% fetal calf serum and 0.1% of saponin.

These antibodies were revealed with secondary antibodies goat anti-mouse IgG (Sigma, #SAB4600042) and goat anti-chicken IgG (Sigma, # SAB4600079) coupled with an Alexa Fluor at the dilution 1/400 in PBS containing 1% FCS, 0.1% saponin (1 hour at RT).

For each condition, 30 pictures/well (representative of the whole well area) were automatically taken using ImageXpress (Molecular Devices) at 20x magnification. All images were generated by ImageXpress® using the same acquisition parameters. From images acquisition, analyses were directly and automatically performed by Custom Module Editor® (Molecular Devices). The following analysis were performed with Custom Module Editor® (Molecular Devices):

- Total neuron survival (number of MAP-2 positive neuron)

- Total neurite network (length of MAP-2 positive neurite in μm)

- Hyperphosphorylation of Tau in neuron (AT100, overlapping Tau/MAP-2 expressed in area, $μm^2$)

## Cell preparation, Aβ$_{1-42}$ exposure, immunostaining and TNF-α quantification in the chronic model of AD

Cortical neurons were prepared as described above for the Aβ$_{1-42}$ acute exposure. STL-101 treatment, Aβ$_{1-42}$ exposure and immunostaining were also performed as described for the acute model with the following modifications:

- Aβ$_{1-42}$ solution was diluted in culture medium at 5μM (0.5μM oligomers, AβO) in the presence of STL-101 for 72 hours

- STL-101 was tested at 0.03, 0.1, 0.3, 1 and 3μM

- AT-100 was used at 1/400

In order to assess the activation of microglia in the cell culture, the level of TNF-α was quantified in cell culture supernatant after 72h (end of the culture) by ELISA according to manufacturer instructions (Abcam, #ab46070).

## Blood samples collection and LC-MS/MS analysis

Blood samples were collected by Biotech Bank (in partnership with the Biological Resources Center of the teaching hospital located in Angers, France) from 160 neurologically healthy and fasting subject (80 women/80 men) from 50 to 90 years old in EDTA tubes (Vacutest with K2 EDTA). Samples were centrifuged for 10 min (4000 rpm at RT) and plasma was aliquoted and stored at -80˚C until analysis. Queuine being an endogenous compound in plasma, surrogate blank matrix (PBS +2% BSA) was used for preparation of calibration standards. 10μL of a

stable labeled internal standard [heavy queuine labelled with 3 $^{15}$N (Synthenova SAS)] was added at 0.2µg/mL to 50µL of plasma before protein precipitation with acetonitrile (MeCN) (150µL) using a phospholipid removal plate. Samples were then centrifuged for 5 min at 14 000 rpm and 4˚C. 150µL of the supernatant was then transferred on PLR 96WP (Phree 30mg/ well part No 8E-S133-TGB Phenomenex) and centrifuge at 3500 rpm for 5 min at 4˚C and then injected in LC-MS/MS. The mass spectrometer was operated in positive ionization. The chromatographic selectivity was carried out by gradient elution using MeCN and 10mM NH4OAc as mobile phases on a Hilic analytical column [ZIC-HILIC sequant 150*2.2mm, 5µm column, SIL-20AC (Shimadzu) automatic sampler maintained at 4˚C, LC-20AD pump (Shimadzu)]. Ionization and detection of queuine and its internal standard was performed on a triple quadrupole mass spectrometer Sciex API 5500, operating in the multiple reaction monitoring and positive ionization mode. Quantitation was done to monitor protonated precursor > product ion transition of m/z 278.1>162.2 for queuine and 281.2>166.2 for its internal standard.

## Chromatographic conditions

Analytical column: Hilic mode
Injection volume: 2µL
Mobile phase A: 10mM NH4OAc
Mobile phase B: MeCN

| Time (min) | Flow rate (mL/min) | % Phase A | % Phase B |
|:---:|:---:|:---:|:---:|
| 0 | 0.2 | 60 | 40 |
| 2 | 0.2 | 90 | 10 |
| 3 | 0.2 | 90 | 10 |
| 3.1 | 0.2 | 60 | 40 |
| 7 | 0.2 | 60 | 40 |
| | | | 55 |

## Mass spectrometry conditions

| MS detector | API 5500 (AB Sciex) |
|:---|:---|
| Ionisation and Acquisition mode | ESI (+), MRM |

| Analyte | Parent (m/z) | Daughter (m/z) | Dwell (msec) | DP (V) | CE (eV) | CXP (V) |
|:---|:---|:---|:---|:---|:---|:---|
| Queuine | 278.1 | 163.23 | 150 | 100 | 21 | 15 |
| Internal standard | 281.2 | 166.2 | 100 | 100 | 21 | 13 |

## Statistical analysis

Data are expressed as the mean and standard error (SEM) as indicated in the figure legends. Methods used to calculate significant differences between groups are indicated in the figure's

legend and were performed using GraphPad Prism. Asterisks indicate statistically significant differences. A $p$-value $< 0.05$ was considered statistically significant.

## Supporting information

**S1 Fig. STL-101 reduces DA neurons loss and α-syn aggregation after MPP+ intoxication.** **(A)** Rat DA neurons were co-treated with STL-101 at the indicated concentrations and MPP + at 4μM for 48h. DA neuron survival was then assessed by TH+ neurons counting. **(B)** Quantification of aggregated α-syn signal obtained with the 2642S antibody in the surviving DA neurons analyzed in A. Control (Ctrl) = culture medium (vehicle). BDNF (Brain-derived neurotrophic factor) was pre-incubated 1h before MPP+ exposure and used as positive control at 50ng/mL. Values are expressed as mean +/- SEM (n = 4–6 culture wells per condition). Statistical analysis was calculated using one-way ANOVA followed by a Dunnett's multiple comparison test (**$p<0.01$, ***$p<0.001$, ****$p<0.0001$ in comparison to MPP+ alone).
(EPS)

**S2 Fig. STL-101 protects against chronic injury of rat cortical neurons with Aβ$_{1-42}$.** **(A)** Quantification of neurite network after Aβ$_{1-42}$ (5μM) injury and STL-101 pre-treatment at the indicated concentrations. **(B)** p-tau quantification in neurites after Aβ$_{1-42}$ injury. **(C)** Neuron survival after Aβ$_{1-42}$ injury and STL-101 pretreatment. **(D)** Measurement of TNF-α release after Aβ$_{1-42}$ injury and STL-101 pretreatment. BDNF was used as a positive control at 50ng/mL. Control (Ctrl) = culture medium with 0.1% DMSO. Results are expressed as a percentage of control condition and mean +/- SEM (n = 4–6) is shown. *$p<0.05$, **$p<0.01$, ***$p<0.001$, ****$p<0.0001$ (in comparison with Aβ$_{1-42}$ alone, One-way ANOVA followed by Dunnett's multiple comparison test).
(EPS)

**S3 Fig. The level of queuine is lower in female mice than male.** Measurement of queuine level was performed by LC-MS/MS in plasma of 6 female and 8 male mice of 45 weeks of age. ****$p<0.0001$ (unpaired t-test).
(EPS)

**S1 Appendix. STL-101 spectra and chromatograms.**
(PDF)

## Acknowledgments

We would like to thank the Stellate Therapeutics team for their work, support and critical advice and discussions. We thank Motac Neuroscience Ltd. for the work with the *in vitro* model of synucleinopathy, Encefa for the work with the MPP+ (10μM) model, Neuro-Sys SAS for the Aβ$_{1-42}$ and MPP+ (4μM) models and Eurofins ADME Bioanalyses for performing the LC-MS/MS analysis of queuine.

## Author Contributions

**Conceptualization:** Patricia Richard, Antoine Danchin, Xavier Manière.

**Formal analysis:** Patricia Richard, Lucie Kozlowski, Xavier Manière.

**Methodology:** Hélène Guillorit.

**Project administration:** Lucie Kozlowski, Patrice Garnier, Nicole C. McKnight, Xavier Manière.

**Resources:** Hélène Guillorit, Antoine Danchin.

**Supervision:** Patricia Richard, Patrice Garnier, Xavier Manière.

**Writing – original draft:** Patricia Richard.

**Writing – review & editing:** Patricia Richard, Nicole C. McKnight, Antoine Danchin, Xavier Manière.

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
