## [Decision Letter · Decision Letter 0]

19 Apr 2021

PONE-D-21-03739

Queuine, a bacterial derived hypermodified nucleobase, shows protection in in vitro models of neurodegeneration

PLOS ONE

Dear Dr. McKnight,

Thank you for submitting your manuscript to PLOS ONE. After careful consideration, we feel that it has merit but does not fully meet PLOS ONE’s publication criteria as it currently stands. Therefore, we invite you to submit a revised version of the manuscript that addresses the points raised during the review process.

1) Reviewer #2 said that not all data in your study has been made available. Please address this concern.

2) Both reviewers are very positive about your manuscript.

3) Reviewer #2 has three major concerns relating to the synthesis of queuine, controls, and the experiments with MPP+. Please address these issues.

4) Reviewer #2 has six minor issues that should be addressed 

5) Please address my editorial points

Overall, the reviews are very positive. Addressing these various issues will improve your study.

We look forward to receiving your revised manuscript.

Kind regards,

Stephan N. Witt, Ph.D.

Academic Editor

PLOS ONE

Journal Requirements:

2. To comply with PLOS ONE submissions requirements, please ensure you have provide methods of sacrifice in the Methods section of your manuscript for all experiments.

'The author(s) received no specific funding for this work.'

We note that one or more of the authors are employed by a commercial company: Stellate Therapeutics

4. Please include a caption for figure 3.

Additional Editor Comments:

Both reviewers liked your manuscript. Reviewer #1 said, "It represents an important contribution to the field and will have a noticeable impact." Reviewer #2 said, "This manuscript has the potential to attract wide-spread readership." I have served as a academic editor for PlOS One for many years, and this is the first time I have seen both reviewers so positive about a manuscript and its potential impact. I strongly urge you to improve the manuscript by addressing the reviewer concerns.

I have some comments that I also urge you to address:

1) Reviewer #2 said "...add a cartoon to describe what you are referring to when you state that queuine replaces guanine at the wobble position 34 of tRNAs with GUN anticodons." The Reader should see what the structure of queuine is and how it "replaces guanine at the wobble position..."

2) Synthetic queuine. (a) You should show a chromatogram of the compound. (b) What is its percent purity? (c) You must have performed mass spec on the synthesized queuine. What was the synthetic compound's mass as determined by mass spec? What is its theoretical mass?

Reviewers' comments:

Reviewer's Responses to Questions

**Comments to the Author**

1. Is the manuscript technically sound, and do the data support the conclusions?

Reviewer #1: Yes

Reviewer #2: Yes

2. Has the statistical analysis been performed appropriately and rigorously? 

Reviewer #1: Yes

Reviewer #2: Yes

3. Have the authors made all data underlying the findings in their manuscript fully available?

Reviewer #1: Yes

Reviewer #2: No

4. Is the manuscript presented in an intelligible fashion and written in standard English?

Reviewer #1: Yes

Reviewer #2: Yes

5. Review Comments to the Author

Reviewer #1: This is an interesting and important study reporting a set of intriguing data on the effect of a bacterial derived hypermodified nucleobase queuine on aggregation potential of proteins associated with neurodegenerative diseases. The authors utilized two in vitro model of synucleinopathy (primary mouse cortical neurons exposed to the pre-formed alpha-synuclein (α-syn) fibrils (huPFFs) and rat DA neurons treated with MPP+ (1-methyl-4-phenylpyridinium)) and an acute Aβ1-42 oligomer injury model of AD in primary cultures of rat cortical neurons and microglia. The authors also showed that in healthy humans, queuine levels are not age-dependent but higher in women. The manuscript is well-written and concise. It represents an important contribution to the field and will have a noticeable impact.

Reviewer #2: This is an interesting manuscript describing a chemical that represents a bacterial product found in the gut of humans responsible for modifying a nucleobase. Here, the authors use a chemically synthesized form of this product to show that varying levels of this chemical are associated with diseases of protein misfolding.

Major comments:

1. The manuscript has very little information about the synthesis of STL-101 (results, methods or figure legends). Please describe this somewhere. For example, the referenced paper for synthesis states that additional analogues of queuine are possible. Is this an analog or the original? Was this compound dissolved in water or a solvent? If a solvent is used, which one and at what final concentration in the experiments?

2. As a related comment, there are no solvent controls mentioned in the manuscript that I can find. Please include in Figure 1D and in Figure 2B-D, and Figure S1 A-D. This will serve as a negative control and as a comparison to the activity observed vs. the neuronal incitor in each of these experiments.

3. The concentration of STL-101 required for significant protection of MPP+ protection of DA neurons was much higher than that for huPFFs (1 uM vs. 0.1 uM) or the Abeta acute model. The abstract describes the utility of queuine for the facilitation of protein misfolding and prevention of protein aggregation. This concept should be further described within your dataset and compared with the MPP+ model which is certainly not considered a traditional protein misfolding model (and required much higher amounts of STL-101 for efficacy).

Minor comments:

1. This manuscript has the potential to attract wide-spread readership. However, many readers will not be familiar with GUN anticodons and wobble position replacement. Therefore, as part of the introduction, could you please add a cartoon to describe what you are referring to when you state that queuine replaces guanine at the wobble position 34 of tRNAs with GUN anticodons?

2. Similarly, I muddled through reading the terms queuine vs. queosine. Please add something of emphasis to the writing (or in an introductory figure) to help the time-crunched reader remember the difference the first time they read it.

3. What program was used to calculate statistics? This should be included in the Stats section.

4. In figure 1C, these data are described in micromolar. However, in figure 1B (legend and results section), the concentration here is described in nanomolar. While the conversion is easy enough, it is inconsistent. Additionally, in Figure 1C, there is a 0 micromolar data point. What was the solvent?

5. Similarly, in Figure 2, there is a discrepancy between nanomolar and micromolar terminology.

6. In the first sentence of the discussion, the word “remarkable” seems a bit overboard and could probably be eliminated without changing the sentence or meaning of your nice work, at all. Moreover, stating in this same sentence that queuine is exclusively produced by bacteria and causes neurodegeneration seems misleading, since, in this study, the STL-101 was made in a lab. Perhaps you could break these thoughts into two different sentences to remind the reader that naturally, queuine is exclusively made by bacteria, but in this study it was chemically synthesized.

6. PLOS authors have the option to publish the peer review history of their article (what does this mean?). If published, this will include your full peer review and any attached files.

Reviewer #1: **Yes: **Vladimir N. Uversky

Reviewer #2: No

---

## [Author Response · Author response to Decision Letter 0]

23 May 2021

Response to the academic editor and reviewers

The academic editor and reviewers’ comments highlighted several important points that needed further clarification which we have provided. We found these comments valuable and below we describe our response point by point. Academic editor and reviewers’ comments are in blue.

Academic editor:

As recommended, we followed PLOS ONE’s style requirements.

2. To comply with PLOS ONE submissions requirements, please ensure you have provided methods of sacrifice in the Methods section of your manuscript for all experiments.

Methods of sacrifice have been added in the Materials and Methods section for each experiment.

3. Please include both an updated Funding Statement and Competing Interests Statement in your cover letter.

Updated Funding and Competing Interests Statements have been included in the cover letter:

“The authors received no specific funding for this work. Authors [PR, LK, HG, PG, NCM, XM] are affiliated to Stellate Therapeutics. The funder provided support in the form of salaries for authors [PR, LK, HG, NCM, XM], but did not have any additional role in the study design, data collection and analysis, decision to publish, or preparation of the manuscript. The specific roles of these authors are articulated in the ‘author contributions’ section.

This affiliation does not alter our adherence to PLOS ONE policies on sharing data and materials.”

4. Please include a caption for figure 3

A caption for Figure 3 (now Figure 4) already existed and can be found on page 12 of the manuscript.

Additional Editor Comments:

1. Reviewer #2 said "...add a cartoon to describe what you are referring to when you state that queuine replaces guanine at the wobble position 34 of tRNAs with GUN anticodons." The Reader should see what the structure of queuine is and how it "replaces guanine at the wobble position..."

We added a figure (Figure 1) that is now part of the introduction explaining the exchange of guanine by queuine into tRNAs and showing the structure of queuine.

2. Synthetic queuine. (a) You should show a chromatogram of the compound. (b) What is its percent purity? (c) You must have performed mass spec on the synthesized queuine. What was the synthetic compound's mass as determined by mass spec? What is its theoretical mass?

Additional details about STL-101 have been added to the STL-101 synthesis paragraph in the Materials and Methods section on page 17 (see response to the major comment #1 of reviewer #1 below).

HPLC, IR, NMR and LC-MS spectra can be found in supporting information as the “queuine dihydrochloride analyses” file.

Reviewer #1: 

Major comments:

1. The manuscript has very little information about the synthesis of STL-101 (results, methods or figure legends). Please describe this somewhere. For example, the referenced paper for synthesis states that additional analogues of queuine are possible. Is this an analog or the original? Was this compound dissolved in water or a solvent? If a solvent is used, which one and at what final concentration in the experiments?

Spectra of STL-101 analyses can now be found as supporting information and the paragraph “STL-101 synthesis” in the Materials and Methods page 17 now indicates the required information about synthesized queuine (STL-101): 

“STL-101 is queuine dihydrochloride (C12H15N5O3 (2HCl)) with a molecular weight (MW) of 350.20 g.mol-1 (natural queuine has a MW of 277.28 g.mol-1). STL-101 purity was determined at 99% by HPLC and was solubilized in water, PBS or DMSO at 10-30 mM as stock solution. Spectra related to STL-101 analysis can be found in the supporting information file entitled “queuine dihydrochloride analyses”.

The final concentration of STL-101 used is indicated on each graph.

2. As a related comment, there are no solvent controls mentioned in the manuscript that I can find. Please include in Figure 1D and in Figure 2B-D, and Figure S1 A-D. This will serve as a negative control and as a comparison to the activity observed vs. the neuronal incitor in each of these experiments.

The vehicle used in each experiment is now indicated in Materials and Methods and in Figure captions. 

For Figure 1A-C (now Fig 2A-C), the vehicle used is PBS (now indicated on page 17/18 of the “Primary culture of cortical neurons and huPFFs challenge” section and Fig 2C caption on page 9).

For Figure 1D (now Fig 2D), culture medium was used as indicated on page 19 in the “Culture of mesencephalic dopaminergic neurons, STL-101 pretreatment and MPP+ injury at 10μM and immunostaining” section and in the figure caption of Fig 2D on page 9.

For Figure 2B-D (now Fig 3B-D) and S1A-D (now Fig S2A-D), STL-101 has been solubilized in DMSO and diluted in culture medium (final concentration in DMSO = 0.1%). The vehicle used for control (Ctrl) = culture medium with 0.1% DMSO as indicated in Figure 3D (page 11) and S2D (page 33) captions. This information can now also be found on page 22 of the Materials and Methods section.

For Figure S1, vehicle = culture medium as indicated in Fig S1B caption (page 33) and on page 19 of the “Culture of mesencephalic dopaminergic neurons, STL-101 and MPP+ (4μM) co-treatment, immunostaining and analysis” section of the Materials and Methods.

3. The concentration of STL-101 required for significant protection of MPP+ protection of DA neurons was much higher than that for huPFFs (1 uM vs. 0.1 uM) or the Abeta acute model. The abstract describes the utility of queuine for the facilitation of protein misfolding and prevention of protein aggregation. This concept should be further described within your dataset and compared with the MPP+ model which is certainly not considered a traditional protein misfolding model (and required much higher amounts of STL-101 for efficacy).

We performed another in vitro experiment using an MPP+ injury at 4μM instead of 10μM and a co-treatment with STL-101 instead of a pre-treatment of 1 day. The data can now be found in Figure S1. We obtained a similar neuroprotective effect of STL-101 (although a significant effect is seen at 3μM only probably due to the lack of pretreatment) in this model than the one previously observed (Fig 2D). In the same model, we also looked at the α-syn signal by IF since it has previously been reported that MPP+ also triggers α-syn aggregation. Indeed Callizot et al., 2019 showed that increased signal of α-syn in DA neurons corresponds to hyper-phosphorylated/aggregated α-syn. Our result (Fig S1B) shows a significant decrease in the α-syn signal in DA neurons when the cells are treated with STL-101 at 1μM and above. 

In order to fully address comment #3, we added the following text in the discussion on page 13 of the manuscript:

“We note that STL-101 concentration required for significant protection is higher with the MPP+ model than the other models (1µM vs. 0.1-0.3µM respectively). STL-101 was added 4 days prior huPFFs exposure, which might contribute to a better protection than no pretreatment or 1 day pretreatment before injury (MPP+ or Aβ1-42). It is also possible that the mitochondrial stress induced by MPP+ alters STL-101 function (e.g., oxidation of the molecule, reduction of cellular intake) which would require higher amount of the molecule to reach protection. Interestingly, significant decrease of α-syn aggregation in the MPP+ model is seen when STL-101 is added without pretreatment at 1µM and above while significant neuronal survival is seen with STL-101 at 3µM. This indicates that some MPP+-induced phenotypes can be rescued by STL-101 near physiological concentrations (α-syn aggregation) while others require STL-101 at pharmacological concentrations (cell survival). This could also indicate that STL-101 treatment reduces/prevents α-syn aggregation before affecting cell survival. It is possible that by optimizing mRNA translation and preventing misfolding of newly synthesized protein, STL-101 delays and/or prevents protein aggregation (in a seeding model such as huPFFs as well as in a model where aggregation might be triggered by mitochondrial dysfunction (e.g., MPP+ intoxication)) that needs to reach a certain level before it can prevent neuronal loss.”

Minor comments:

1. This manuscript has the potential to attract wide-spread readership. However, many readers will not be familiar with GUN anticodons and wobble position replacement. Therefore, as part of the introduction, could you please add a cartoon to describe what you are referring to when you state that queuine replaces guanine at the wobble position 34 of tRNAs with GUN anticodons? 

We have now added a Figure (Figure 1) in the introduction describing queuine integration into tRNAs.

2. Similarly, I muddled through reading the terms queuine vs. queosine. Please add something of emphasis to the writing (or in an introductory figure) to help the time-crunched reader remember the difference the first time they read it.

Figure 1 also states what is queuine and what is queuosine.

3. What program was used to calculate statistics? This should be included in the Stats section.

GraphPad Prism was used to perform all the stats and this information is now in the “Statistical analysis” section on page 26.

4. In figure 1C, these data are described in micromolar. However, in figure 1B (legend and results section), the concentration here is described in nanomolar. While the conversion is easy enough, it is inconsistent. Additionally, in Figure 1C, there is a 0 micromolar data point. What was the solvent?

The concentration has been changed to µM in the caption of Figure 2B (former Figure 1B). As now indicated in the caption of Figure 2C, the solvent is PBS.

5. Similarly, in Figure 2, there is a discrepancy between nanomolar and micromolar terminology.

µM is now used in the caption of Figure 3 (former Figure 2).

6. In the first sentence of the discussion, the word “remarkable” seems a bit overboard and could probably be eliminated without changing the sentence or meaning of your nice work, at all. Moreover, stating in this same sentence that queuine is exclusively produced by bacteria and causes neurodegeneration seems misleading, since, in this study, the STL-101 was made in a lab. Perhaps you could break these thoughts into two different sentences to remind the reader that naturally, queuine is exclusively made by bacteria, but in this study it was chemically synthesized

As suggested, the first sentence of the discussion is now:

“Here we have shown protective properties of chemically synthesized queuine, a molecule naturally exclusively produced by bacteria, in several in vitro models of proteinopathies and neurodegeneration.”

---

## [Decision Letter · Decision Letter 1]

31 May 2021

Queuine, a bacterial-derived hypermodified nucleobase, shows protection in in vitro models of neurodegeneration

PONE-D-21-03739R1

Dear Dr. Richard,

We’re pleased to inform you that your manuscript has been judged scientifically suitable for publication and will be formally accepted for publication once it meets all outstanding technical requirements.

Kind regards,

Stephan N. Witt, Ph.D.

Academic Editor

PLOS ONE

Additional Editor Comments (optional):

Reviewers' comments:

Reviewer's Responses to Questions

**Comments to the Author**

1. If the authors have adequately addressed your comments raised in a previous round of review and you feel that this manuscript is now acceptable for publication, you may indicate that here to bypass the “Comments to the Author” section, enter your conflict of interest statement in the “Confidential to Editor” section, and submit your "Accept" recommendation.

Reviewer #2: All comments have been addressed

2. Is the manuscript technically sound, and do the data support the conclusions?

Reviewer #2: (No Response)

3. Has the statistical analysis been performed appropriately and rigorously? 

Reviewer #2: (No Response)

4. Have the authors made all data underlying the findings in their manuscript fully available?

Reviewer #2: (No Response)

5. Is the manuscript presented in an intelligible fashion and written in standard English?

Reviewer #2: (No Response)

6. Review Comments to the Author

Reviewer #2: (No Response)

7. PLOS authors have the option to publish the peer review history of their article (what does this mean?). If published, this will include your full peer review and any attached files.

Reviewer #2: No

---

## [Editor Report · Acceptance letter]

29 Jul 2021

PONE-D-21-03739R1 

Queuine, a bacterial-derived hypermodified nucleobase, shows protection in *in vitro* models of neurodegeneration 

Dear Dr. Richard:

I'm pleased to inform you that your manuscript has been deemed suitable for publication in PLOS ONE. Congratulations! Your manuscript is now with our production department. 

Kind regards, 

on behalf of

Dr. Stephan N. Witt 

Academic Editor

PLOS ONE